# Interaction of *Pelargonium sidoides* Compounds with Lactoferrin and SARS-CoV-2: Insights from Molecular Simulations

**DOI:** 10.3390/ijerph19095254

**Published:** 2022-04-26

**Authors:** Federico Iacovelli, Gaetana Costanza, Alice Romeo, Terenzio Cosio, Caterina Lanna, Antonino Bagnulo, Umberto Di Maio, Alice Sbardella, Roberta Gaziano, Sandro Grelli, Ettore Squillaci, Alessandro Miani, Prisco Piscitelli, Luca Bianchi, Mattia Falconi, Elena Campione

**Affiliations:** 1Structural Bioinformatics Group, Department of Biology, University of Rome “Tor Vergata”, 00133 Rome, Italy; federico.iacovelli@uniroma2.it (F.I.); alice.romeo@uniroma2.it (A.R.); falconi@uniroma2.it (M.F.); 2Virology Unit, Department of Experimental Medicine, Tor Vergata University Hospital, 00133 Rome, Italy; costanza@med.uniroma2.it (G.C.); grelli@med.uniroma2.it (S.G.); 3PhD Course in Microbiology, Immunology, Infectious Diseases, and Transplants (MIMIT), Department of Experimental Medicine, University of Rome Tor Vergata, 00133 Rome, Italy; terenziocosio@gmail.com (T.C.); roberta.gaziano@uniroma2.it (R.G.); 4Dermatology Unit, Department of Systems Medicine, Tor Vergata University Hospital, 00133 Rome, Italy; caterinalanna.cl@gmail.com (C.L.); sbardellaalice@gmail.com (A.S.); luca.bianchi@uniroma2.it (L.B.); 5NEILOS SRL, 80063 Piano di Sorrento, Italy; a.bagnulo@neilos2015.com (A.B.); u.dimaio@neilos2015.com (U.D.M.); 6Department of Biomedicine and Prevention, University of Rome “Tor Vergata”, 00133 Rome, Italy; ettore.squillaci@uniroma2.it; 7Department of Environmental Sciences and Policy, University of Milan, 20133 Milan, Italy; alessandro.miani@unimi.it; 8Health Education and Sustainable Development, University of Naples Federico II, 80125 Naples, Italy; priscofreedom@hotmail.com

**Keywords:** molecular docking, molecular dynamics, MM/GBSA, *Pelargonium sidoides*, lactoferrin, SARS-CoV-2

## Abstract

(1) Background: *Pelargonium sidoides* extracts and lactoferrin are two important natural, anti-inflammatory, and antiviral agents, which can interfere with the early stages of SARS-CoV-2 infection. Molecular docking and molecular dynamics simulation approaches have been applied to check for the occurrence of interactions of the *Pelargonium sidoides* compounds with lactoferrin and with SARS-CoV-2 components. (2) Methods: Computational methods have been applied to confirm the hypothesis of a direct interaction between *PEL* compounds and the lactoferrin protein and between *Pelargonium sidoides* compounds and SARS-CoV-2 Spike, 3CLPro, RdRp proteins, and membrane. Selected high-score complexes were structurally investigated through classical molecular dynamics simulation, while the interaction energies were evaluated using the molecular mechanics energies combined with generalized Born and surface area continuum solvation method. (3) Results: Computational analyses suggested that *Pelargonium sidoides* extracts can interact with lactoferrin without altering its structural and dynamical properties. Furthermore, *Pelargonium sidoides* compounds should have the ability to interfere with the Spike glycoprotein, the 3CLPro, and the lipid membrane, probably affecting the functional properties of the proteins inserted in the double layer. (4) Conclusion: Our findings suggest that *Pelargonium sidoides* may interfere with the mechanism of infection of SARS-CoV-2, especially in the early stages.

## 1. Introduction

Due to the emergence of pandemic diffusion of SARS-CoV-2, health care systems and emergency medical services have become overwhelmed [1]. Some measures, such as lock-down of communities, social distancing, and quarantine-type for those suspected to be infected, can, at least in part, slow the COVID-19 (*COronaVIrus Disease 19*) spread [2] and therefore enable the health systems to cope. In the worldwide search for a response to the COVID-19 pandemic, different natural remedies against COVID-19 have been reported [3,4,5]. Among these, *Pelargonium sidoides* and lactoferrin showed important anti-inflammatory, anti-oxidative, and antiviral properties, administered alone or in combination [3,4,5,6,7,8].

*Pelargonium sidoides* (*PEL*; Geraniaceae) is an African medicinal plant, traditionally used for curing different diseases, including diarrhea, colic, gastritis, tuberculosis, cough, hepatic disorders, menstrual complaints, and gonorrhea [9]. The common name, *umckaloabo*, represents the Zulu (“*Umkuhlune*”—coughing and fever; and “*Uhlabo*” = pain in the chest) [10] word describing ‘severe cough’. Indeed, its extracts are successfully employed in modern phytotherapy in Europe to cure infectious diseases of the respiratory tract [5,8,9].

*Pelargonium sidoides* is indicated for the common cold [11], cough, and bronchitis [12]. *Pelargonium sidoides* root extracts preparations are available in some European Countries with a full marketing authorization (e.g., Bulgaria, Czech Republic, Germany) or registered as a traditional herbal medicinal product (e.g., Austria, Hungary, Italy, The Netherlands, Poland, Spain, Sweden), and are widely used for acute bronchitis and other respiratory infections [11]. Relevant key metabolites assumed to be active are hydrolysable tannins, catechins, gallic acid, and methyl gallate, including some unusual O-galloyl-C-glucosyl flavones, scopoletin, 6,8-dihydroxy-5,7-dimethoxycoumarin, 5,6,7-trimethoxycoumarin. Other coumarins, as well as, quercetin 3-O-b-D-glucoside, myricetin, and other flavonoids, have been isolated. This herbal medicine has been experimentally proven for anti-viral activity as reported for *Pelargonium sidoides* that interfere in vitro with the replication of different respiratory viruses, including human coronaviruses [13], influenza virus (in vitro and in vivo) [14], and Rhinovirus isolated from patients with severe asthma [15], by stimulating IFN-b in vitro, while gallic acid enhanced the expression of iNOS and TNF-α [16].

Lactoferrin (Lf) is a glycoprotein of the transferrin family [17,18], synthetized by exocrine glands and neutrophils and is present in human milk and in all secretions [17,18]. This protein is one of the most important factors of innate immunity, constituting a barrier against pathogens colonizing both mother and fetal habitats [19], and it was demonstrated that it could also act as a potential nutraceutical capable of contrasting SARS-CoV-2 infection [6,7].

Lactoferrin has four essential activities: chelation of two ferric ions per molecule, interaction with anionic compounds, translocation into the nucleus, and modulation of inflammation and iron homeostasis [3,17,18]. Lf’s capability to chelate two ferric ions per molecule influences bacterial and viral replication and hinders reactive oxygen species formation (ROS) [17,20,21]. The binding of Lf to anionic surface components, thanks to its cationic features, is associated with the host protection against bacterial and viral adhesion and entry [17]. Moreover, the entrance of Lf into host cells, and its translocation into the nucleus [22,23], is related to its anti-inflammatory function [24]. In a recent study, we demonstrated that bovine Lf (bLf) exerts its antiviral activity either by direct binding to the SARS-CoV-2 particles or by obscuring their host cell receptors [6]. Moreover, the results obtained through the computational approaches strongly supported the hypothesis of a direct recognition between the bLf and the spike glycoprotein [6].

As reported in literature [5], the combination *Pelargonium sidoides* + Lf (*PEL*IRGOSTIM), can reduce in vitro the release of proinflammatory cytokines, oxidants, and bacteria growth, most likely preventing leukocyte chemiotaxis with a reduced inflammatory pattern. *PEL* and Lf used alone were able to reduce LPS-induced proinflammatory IL-1β, as well as reduce ROS, nitrite, and bacteria growth. It can be hypothesized that this synergistic effect may counteract SARS-CoV-2 infection.

In view of this context, it is important to understand if the compounds contained in *PEL* extracts can interfere with the lactoferrin structure and function and what types of interactions can be established with the key components of SARS-CoV-2.

To this aim, in this study we applied computational methods to verify the hypothesis of a direct interaction between *PEL* compounds and the Lf protein and with SARS-CoV-2 Spike, 3CLPro, RdRp proteins, and membrane. Selected high-score complexes obtained with molecular docking were structurally investigated through classical molecular dynamics (MD) simulations, rescoring their interaction energies using the molecular mechanics energies combined with generalized Born and surface area continuum solvation (MM/GBSA) method [25]. The results obtained from the computational analyses suggest that *PEL* compounds and Lf could synergistically interfere with the mechanism of infection of SARS-CoV-2, especially in the early stages.

## 2. Materials and Methods

### 2.1. Molecular Docking of the PEL Compounds

The structure of the most relevant compounds present in the *PEL* root extracts (Appendix A), including 6-7-8-trihydroxycoumarin, 5-6-7-trimethoxycoumarin, 6-8-dihydroxy-7-methoxycoumarin, apocynin, 7-acetoxy-5-6-dimethoxycoumarin, artelin, dihydroxybenzoic-acid, caffeic acid, ferulic acid, epigallocatechin-3-gallate, dimethoxycoumarin, gallic acid, fraxetin, isofraxoside, homovanillic acid, gallocatechin, magnolioside, isovitexin, isoorientin, pentagalloyl-glucose, taxifolin-3-glucoside, scopoletin, quercetin, vanillic acid, umckalin, and vitexin have been downloaded from the PubChem database in the SDF format and converted in MOL2 using the openBabel routines [26]. The structures of the bovine lactoferrin (ID: 1BLF) [27], spike glycoprotein [28], 3CL protease (ID: 6LU7) [29] and catalytic subunit of the RdRp polymerase (ID: 7BV2) [30] have been downloaded from the PDB database. Protein and drug structure files have been converted into pdbqt format using the prepare_receptor4.py and the prepare_ligand4.py tools of the AutoDockTools4 software [31]. The molecular docking simulations have been performed using the AutoDock Vina 1.2.0 program [32]. AutoDock Vina is one of the fastest and most widely used open-source programs for molecular docking, which has been recently updated introducing new docking methods and an expanded force field [32]. One docking simulation, including ten docking runs, has been carried out for each drug. For each docking simulation, the AutoDock Vina program selects 10 binding poses representing the cluster centroids of all the different conformations, generated in each run using a Lamarckian Genetic Algorithm coupled to a gradient optimization algorithm. For the lactoferrin, the simulation box has been placed to include the whole protein structure. A box of size x = 23.25 Å, y = 24.38 Å, z = 25.88 Å has been placed over the HR1 internal region of the spike glycoprotein A monomer. A box of size x = 22.75 Å, y = 23.75 Å, z = 24.78 Å has been placed over the 3CLpro binding site for located between domains I and II of the protein. Finally, after the removal of the RNA molecule, a box has been set to include the whole RdRp polymerase structure.

### 2.2. Molecular Dynamics of the Best Docking Complexes

The topology and coordinate files of the best docking complexes (7, 12, 7, and 1 for the lactoferrin, Spike, 3CLpro, and RdRp polymerase, respectively) were generated using the tLeap module of the AmberTools program [33], parametrizing the structures through the AMBER ff19SB force field [34]. The structures were placed into a rectangular box, solvated with TIP3P water molecules [35], and neutralized by adding the correct amount of neutralizing counterions, forcing a minimum distance between the structure and the box sides of 12 Å. For each structure, a minimization run of 500 steps using the steepest descent algorithm followed by 1500 steps of a conjugate gradient was performed to remove any unfavorable interactions. The systems were gradually heated from 0 to 300 K in the NVT ensemble over a period of 500 ps using the Langevin thermostat, applying a restraint of 0.5 kcal·mol^−1^·Å^−2^ on each protein and ligand atom to relax the solvent. Through 1.0 ns long equilibration runs, the restraint forces were gradually decreased to 0.1 kcal·mol^−1^·Å^−2^. The systems were simulated using an isobaric-isothermal (NPT) ensemble for 1.0 ns, setting the temperature to 300 K and the pressure to 1.0 atm using the Langevin barostat [36]. The SHAKE algorithm [37,38] was used to constrain the covalent bonds involving hydrogen atoms. The systems were then subjected to a 100.0 ns equilibration production run, with a timestep of 2.0 fs, using the PME method [39] for long-range interactions and a cut-off of 9.0 Å for the short-range interactions. These simulations have been performed using the GPU-enabled version of AMBER 16.0 *pmemd* module on, saving systems coordinates every 1000 steps.

### 2.3. Molecular Dynamics of the SARS-CoV-2 Membrane–PEL Compounds Interaction

Topology and coordinate files for a 230 × 230 Å membrane and the *PEL* compounds have been generated using the Membrane Builder tool of the CHARMM-GUI interface [40]. The CHARMM36m force field for lipids [41] was used to parametrize the membrane system. Membrane composition mimics that of a SARS-CoV-2 viral envelope, including cholesterol (30%), 3-palmitoyl-2-oleoyl-d-glycero-1-phosphatidylcholine (6%), 2,3 dipalmitoyl-d-glycero-1-phosphatidylcholine (4%), 3-palmitoyl-2-oleoyl-d-glycero-1-phosphatidylethanolamine (18%), 2,3 dipalmitoyl-d-glycero-1-phosphatidylethanolamine (12%), 3-palmitoyl-2-oleoyl-d-glycero-1-phosphatidylserine (6%), 2,3 dipalmitoyl-d-glycero-1-phosphatidylserine (4%) and sphingomyelin d18;1/16;0 (20%) [42]. Parameters for the compounds have been generated using the CGenFF program (https://cgenff.umaryland.edu (accessed on 17 April 2022)) and the CHARMM general force field. The compounds-membrane system was inserted in a rectangular box filled with TIP3P water molecules [35] and neutralized with 0.15 M of NaCl ions. The final system included 395,855 atoms. To remove unfavorable interactions, the system has been minimized in ten runs, each including 2000 steps. A constraint of 20.0 kcal/mol was initially applied on each atom, sequentially halved in the subsequent runs, and finally removed in the last run. Minimized systems have been thermalized in a canonical ensemble (NVT) using a timestep of 1.0 fs, gradually increasing the temperature from 0 to 310 K every 30 ps using Langevin dynamics [43] and applying a constraint of 5.0 kcal/mol on protein and membrane atoms. The system was then equilibrated in an anisotropic NPT (NPT-A) ensemble using the Nosè-Hoover Langevin piston method [36,44] and a constant pressure of 1.0 atm, gradually releasing the constraints applied on protein and membrane every 250 ps during a 2250 ps run. Then, the timestep was increased to 2.0 fs and the system was simulated for 150 ns. + Electrostatic interactions have been calculated using the PME method [39], while the cut-off for non-bonded interactions was set to 12.0 Å. This simulation has been performed using the NAMD 2.13 program [45], saving system coordinates every 1000 steps.

### 2.4. Trajectory Analysis

Membrane thickness has been evaluated using the VMD Membrane Analysis Tool [46]. 2D thickness maps have been realized using an in-house Python script. Distance analyses were performed using an in-house VMD Tcl script. The most representative structure of the simulated proteins (clusters centroid) has been extracted through a cluster analysis of the trajectories using the g_cluster tool of Gromacs [47], using a cut-off of 0.16 nm and the *gromos* clustering algorithm [48]. Generalized Born and surface area continuum solvation (MM/GBSA) analysis [25] have been performed over the last 30 ns of all the simulations (excluding the membrane system), using the MMPBSA.py.MPI program implemented in the AMBER16 software [49]. Images have been generated using the VMD [50] Humphrey or Chimera software [51].

## 3. Results

### 3.1. Prediction of PEL Compounds Toxicity

The oral acute toxicity, organ toxicity, immunotoxicity, genetic toxicity endpoints, nuclear receptor signalling, and stress response pathways of PEL compounds have been evaluated using the ProTox-II webserver [52]. Appendix A indicates the oral acute toxicity (LD50) as mg/kg, predicted toxicity classes (I–VI), and the predicted toxicity model for PEL compounds. Among the 26 molecules, only quercetin, isovitexin, and isoorientin are predicted as class III compounds, specified as toxic after swallowing (50 < LD50 ≤ 300). However, in the literature, there is no evidence of toxicity for isovitexin and isoorientin, and for quercetin when administered at up to 5 g daily [53]. The compounds identified as class IV and V could be harmful after swallowing (300 < LD50 ≤ 2000 and 2000 < LD50 ≤ 5000, respectively). Additionally, in this case, from the analysis of the literature, we found no evidence of toxicity for most of them, with only a minor cytotoxic effect of coumarins when administered at doses higher than 5 g [54]. Gallocatechin and epigallocatechin-3-gallate (identified as class VI compounds) toxicity is debated in the scientific community since hepatic failure has been associated with the intake of epigallocatechin-3-gallate [54]. However, the intake of these substances up to 0.9 g per day should be safe, and promoted some decrease in LDL cholesterol [55]. The currently used clinical dose of EPs 7630, the main proprietary Pelargonium sidoides root extract used for curing common respiratory diseases, is ~60 mg [8]. Given the recommended low dosage, the use of this extract should be safe, as toxic concentrations of the compounds would never be reached.

### 3.2. Interaction of PEL Compounds with the Bovine Lactoferrin

The interaction energies from the molecular docking simulations of the *PEL* compounds and the bLf are shown in Table 1. As already reported for molecules like flavonoids and other small polyphenols, the compounds bind in proximity of His91, Leu687, and Thr688 [56], which define the margins of a cavity (Figure 1) with a size sufficient to accommodate even the largest compound, the pentagalloyl glucose. Notably, this cavity is located far away from the SARS-CoV-2 spike glycoprotein putative binding site that we identified in a previous work [6], suggesting that the binding of the compounds do not interfere with the recognition of the viral protein.

The seven compounds (Table 1, grey) showing energies higher than −7.5 kcal/mol have been investigated performing 100 ns long classical molecular dynamics simulation. This procedure allowed us to validate the binding affinities and to verify their effect on the bovine lactoferrin structural properties.

Figure 2A–G shows the results from the RMSD analysis of the seven trajectories. As can be observed, the RMSD values reported as a function of the simulation time indicate that the lactoferrin structure is stable and does not deviate from the reference one. The analysis of the secondary structure elements during the simulation time confirms that the presence of the compounds into the cavity does not alter the lactoferrin structural properties (Appendix A). Finally, the MM-GBSA method allowed us to re-evaluate the interaction energies between the lactoferrin and the seven compounds. In general, the analysis partially overturns the docking results, highlighting marked differences between the various compounds (Table 2), with taxifolin-3-glucoside, isoorientin, gallocatechin, and artelin showing higher interaction energies than pentagalloyl-glucose, epigallocatechin-3-gallate, and vitexin. As a matter of fact, it is reasonable to assume that these compounds can take advantage of the lactoferrin structure as a molecular carrier, without altering its structural and dynamical properties.

### 3.3. Interaction of PEL Compounds with the SARS-CoV-2 Protease (3CLpro)

To check the interaction of *PEL* compounds with the viral protease, we first optimized the protein structure through a 100-ns long classical molecular dynamics simulation, as described in the methods section. After a clustering procedure, we extracted a reference structure to be used as receptor for the molecular docking experiments, selecting as the research area the binding site identified through the X-ray diffraction experiments [29]. As reported in Table 3, the subset of compounds (Figure 3A–G) showing an interaction energy higher than 7.5 kcal/mol was investigated through MD simulations to evaluate if the identified interactions lead to the formation of stable complexes.

Besides, the analyses show that all seven compounds interact with the catalytic dyad of the protease, namely His41 and Cys145 (Figure 4). Finally, the MM-GBSA method allowed us to re-evaluate the interaction energies between the protease and the seven compounds. In general, the analysis confirms the docking results but highlights more marked differences between the various compounds (Table 4), with vitexin, pentagalloylglucose, magnolioside, and isoorientine showing interaction energies higher than those of artelin, quercetin, and isovitexin. Therefore, it is reasonable to assume that of these seven compounds, at least the first four could have an inhibitory activity against the virus protease.

### 3.4. Interaction of PEL Compounds with the SARS-CoV-2 Spike Glycoprotein

The reference structure for docking simulations on the Spike glycoprotein arises from the calculations carried out in our previous work [28]. The molecular docking experiments were carried out on the trimeric structure of the Spike, shown in Figure 5A–G, selecting as the research area the binding site identified in our work and in the literature, namely the HR1 domains, which are responsible for the conformational change that allows the entry of the virus inside cells [28]. Interfering with these domains using compound as fusion inhibitors, it should be possible to block the glycoprotein in its prefusion state before it can enter the cells by recognizing the ACE2 receptor. As reported in Table 5, a large number of compounds, higher than that observed for the protease (highlighted in grey), show energies higher than −7.5 kcal/mol and have been investigated performing 100 ns long classical molecular dynamics simulation.

Appendix A shows the data related to the distance calculated as a function of the simulation time, for eleven out of twelve simulated systems, between the HR1 domains of the Spike glycoprotein and each compound. The lack of artelin in the results is due to the instability of the complex: in fact, this molecule moves away from the pocket already in the equilibration phases of the system. Several equilibration attempts all led to the unbinding of the compound, so it was excluded from the subsequent analyses. As can be observed, the compounds identified by molecular docking remain stably located in the original positions or relocate to it during the simulation time due to a reorganization of the interactions occurred during the equilibration phases. In all cases, except for isovitexin, the compounds interact exclusively with the HR1 domain. Isovitexin, on the other hand, tends to occupy a region between the HR1 domain and the lower part of the cavity generated by the coupling of the three Spike monomers. The MM-GBSA method allowed us to re-evaluate the interaction energies between the Spike glycoprotein and these compounds. In general, the analysis overturns the docking results highlighting marked differences between the various compounds (Table 6), with taxifolin-3-glucoside, pentagalloyl-glucose, magnolioside, isovitexin, epigallocatechin-3-gallate, gallocatechin, and isofraxoside showing higher interaction energies (especially the first three).

### 3.5. Interaction of PEL Compounds with the SARS-CoV-2 RdRp Polymerase

The molecular docking experiments have been carried out, using the catalytic subunit of the RNA-dependent polymerase as a receptor, deposited on the PDB database with code 7BV2 [29]. The RdRp structure has been optimized through a 100-ns long classical molecular dynamics simulation, removing the RNA molecule from the crystal to make all the putative binding sites accessible. After a clustering procedure, we extracted a reference structure that we used as receptor for the molecular docking experiments, selecting the entire subunit as the research area. As can be observed in Table 7, only the pentagalloylglucose showed an interaction energy higher than 7.5 kcal/mol, which we considered the minimum threshold to consider that the observed interactions are not due to chance. This is a predictable result given the difficulty of targeting viral polymerases. This compound binds in proximity of the RNA binding site of the polymerase (Figure 6). However, to better analyze the interaction between pentagalloylglucose and the polymerase, we have performed a 100 ns long molecular dynamics simulation of the complex.

Appendix A shows the distance between the pentagalloyl glucose, and the binding site identified on the viral polymerase as a function of the simulation time. From the point of view of geometric stability, pentagalloylglucose was able to establish molecular interactions sufficient to maintain the original pose in the binding site. However, an evaluation of the interaction energy through the MMGBSA method returned a value of −10.0 kcal/mol, which we do not consider sufficient to validate this interaction, as this compound could easily be displaced by the RNA binding.

### 3.6. Interaction of PEL Compounds with the SARS-CoV-2 Membrane

It is of fundamental importance to verify whether the *PEL* compounds could interact with the membrane of the SARS-CoV-2 virus. In fact, being rich in phenolic acids, *PEL* extracts could interfere with the lipid membrane dynamical properties, therefore affecting the properties of the proteins inserted in the double layer. A model of the membrane, including the solvent and the 27 main molecules present in the extracts of *PEL*, has been created using the Packmol software [57], which allows us to insert a predefined number of molecules within a simulation box in a semi-automatic way (Appendix A). The system has been simulated for 150 ns through accelerated molecular dynamics and has been compared with the same membrane simulated in the absence of the compounds. In Figure 7 it is possible to observe the entry of dihydrobenzoic acid (in cyan), of vanillic (gray) and homovanillic acid (violet), of scopoletin (orange), vitexin (green), isofraxoside (red), and gallocatechin (lilac).

Furthermore, almost all the other compounds tend to contact the surface of the membrane, and while not penetrating inside it, their positions suggest that they can alter its external electrostatic potential and induce a curvature of the lipid bilayer. Specific analyses have been then carried out on the structural parameters of the membrane aimed at demonstrating the differences between the two systems and the effect exerted by the *PEL* molecules. To this aim, we used VMD’s MEMBPLUGIN plugin [46], which allows us to analyze all the types of lipid bilayers. Figure 8 shows the membrane thickness as a function of the simulation time for the system simulated in the absence (black) and in the presence of molecules (red). It is evident how the entry of the molecules can alter the average thickness of the membrane, which increases from about 41 to 43.5–44 Å in the presence of the compounds.

These effects become more evident by projecting the thickness of the membrane on a plane, as shown in Figure 9. In this representation, the average thickness of the membrane, indicated by the color scale that goes from white to dark green, has been plotted as a function of the transverse plane of the membrane itself for the simulated system in the presence (A) and absence of the compounds (B). Through this representation, it can be observed that the increase in thickness is more marked in proximity of the entry sites or surface contact of the compounds, identified by the red circles. However, the effect seems extended to the entire membrane, suggesting that the compounds can deeply interfere with the dynamics of the lipid bilayer, suggesting a possible modulatory effect on the dynamics of the proteins inserted in the membrane.

Finally, an additional analysis was carried out aimed at definitively confirming the effect of the entry of compounds into the membrane, using the VMD Density Profile Tool, an analysis plugin which calculates 1-D projections of various atomic densities [58]. Appendix A show, respectively, the density profiles of the two layers of the membrane simulated in the absence and presence of the phytocomplex compounds. Under normal conditions, the density of the two layers of the membrane is almost perfectly comparable, as can be observed in Appendix A. On the other hand, when *PEL* compounds are present, a shift in the density profile occurs with an increase in density in the outer layer, i.e., the one in contact with the compounds. This shift is directly correlated to an increase in the local curvature of the membrane induced by the contact with the molecules, which is in turn correlated with the modulation of protein functionality [59]. In fact, it has been shown that the increase in curvature facilitates the entry of molecules inside the membranes [60], favoring their biological effect.

## 4. Discussion

In this study, we applied computational methods to check for the occurrence of interactions of the PEL compounds with Lf and SARS-CoV-2 components.

*Pelargonium sidoides* preparations have been trialed clinically for cough, even if the clinical evidence is high only for bronchitis and the common cold. [61,62,63]. *Umckaloabo* preparations are generally considered to be safe, although gastrointestinal discomfort (stomach pain, heartburn, nausea, or diarrhea) might occur [8]. *PEL* extracts are commonly employed in modern phytotherapy in Europe to cure infectious diseases of the respiratory tract [4,5].

On the other hand, lactoferrin has been proven to act as a scavenger against iron overload and inflammation in lung epithelium of mice infected by *Pseudomonas aeruginosa* [64,65] and was found to rebalance lung iron-handling proteins and to decrease broncho-alveolar iron overload, one of the main actors in infection progression and exacerbation [3]. Moreover, several studies described Lf’s antiviral activity towards enveloped and naked viruses, related to different virus families, such as *Retroviridae* (human immunodeficiency viruses), *Papillomavirid*ae (human papillomavirus), *Herpersviridae* (Cytomegalovirus, Herpes simplex virus), Caliciviridae (feline calicivirus), *Flaviviridae* (hepatitis C virus, Japanese encephalitis virus), *Reoviridae* (rotavirus), *Adenoviridae* (adenovirus), *Pneumoviridae* (respiratory syncytial virus), *Paramixoviridae* (parainfluenza virus), *Orthomixoviridae* (influenza A virus), and other viruses [3]. bLf has been found to hinder viral entry into host cells through its competitive binding to the cell surface receptors, mainly negatively charged compounds such as glycosaminoglycans (GAGs) [66,67,68,69,70,71]. In addition, Lf was found to prevent viral infections by binding to dendritic cell-specific intercellular adhesion. Overall, the antiviral effect of Lf occurs in the early phase of infection, preventing the entry of viral particles into the host cells, either by blocking cellular receptors and/or by directly binding to the viral particles. Further, Lf is also able to exert an antiviral activity when it is added in the post-infection phase, as demonstrated in Rotavirus infection by Superti et al., [72] and in HIV infection by Puddu et al. [73].

In a previous study, Terlizzi et al. demonstrated that the combination of *PEL* and Lf could exert additive/synergistic pharmacological activities as anti-inflammatory, antioxidant, and antimicrobial agents compared with the single components [5]. They found that *PEL* and Lf used alone were able to reduce LPS-induced proinflammatory IL-1β, as well as reduce ROS, nitrite, and bacteria growth. More importantly, the combination of *PEL* with Lf showed an additive pharmacological activity in terms of antioxidant and antimicrobial activities. Data demonstrated that the combination of *PEL* + Lf significantly reduced the levels of IL-1β after LPS stimulation. This effect was an innovative and hitherto unknown combination, able to attenuate inflammation-related pathways [3,5,6,7].

Molecular docking and molecular dynamics simulation approaches strongly supported the hypothesis of a direct recognition between the bLf and the SARS-CoV-2 spike glycoprotein [6,28]. The affinity between their molecular surfaces, the large number of atomistic interactions detected, and their persistence during the simulation suggested that this recognition is very likely to occur and that bLf may hinder the spike binding to the ACE2 receptor, thus blocking virus entry into host cells [6].

In this scenario, we have carried out a series of molecular docking and molecular dynamics simulations to identify possible interactions between *PEL* and Lf and between *PEL* and some of the SARS-CoV-2 components.

First, we analyzed by molecular docking if the interaction between *PEL* compounds and Lf could alter its functional properties, hampering the interaction with other macromolecules as Spike. Our results are fully in agreement with literature since it has been demonstrated that the structure and activity of lactoferrin is not altered by the presence of organic molecules or metal ions different from iron [17,18]. Based on these results, the combined use of *PEL* and Lf in a dietary supplement has been acknowledged.

Subsequently, computational studies have been carried out to evaluate a possible interaction between *PEL* compounds and the SARS-CoV-2 3CLpro protein. In a recent work it has been shown that compounds capable of interacting with both residues of the 3CLpro catalytic dyad can inhibit the activity of this enzyme between 50 and 88% [74]. Moreover, other flavonoids have shown inhibitory activity against this protein [75]. In general, the results confirms that several *PEL* compounds can stably interact with the active site of the protease in proximity of the catalytic dyad, suggesting an inhibitory activity against the virus protease.

The interaction of *PEL* compounds with the SARS-CoV-2 Spike glycoprotein has been also investigated. The molecular docking experiments were carried out on the trimeric structure of the Spike, selecting as research area the HR1 domains, which are responsible for the conformational change that allows the entry of the virus inside cells [28]. Interfering with these domains should block the glycoprotein in its prefusion state, before it can enter the cells by recognizing the ACE2 receptor. Based on these results, we could assume that several analyzed compounds could interfere with the conformational changes of the Spike glycoprotein.

The interaction of *PEL* compounds with the SARS-CoV-2 RdRp polymerase has also been checked, but from the results no molecules of the *PEL* extracts may have the ability to interfere with the viral polymerase.

Finally, we verified whether the *PEL* compounds could interact with the membrane of the SARS-CoV-2 virus. In fact, being rich in phenolic acids, *PEL* extracts could interfere with the lipid membrane dynamical properties, consequently affecting the motions of the proteins inserted in the double layer. The MD simulation analyses suggest that the interaction of *PEL* compounds with the membrane can alter its external electrostatic potential, inducing a curvature of the bilayer. However, this effect extends to the entire membrane, suggesting that *PEL* compounds can penetrate the viral lipid bilayer, alter its physical properties, and also interact with viral proteins in infected cells.

## 5. Conclusions

In conclusion, our results suggest that *PEL* and Lf could synergistically act, interfering against SARS-CoV-2 in silico in different ways. This represents an important and necessary first evidence, useful to setting subsequent in vitro and in vivo studies.

## Figures and Tables

**Figure 1 ijerph-19-05254-f001:**
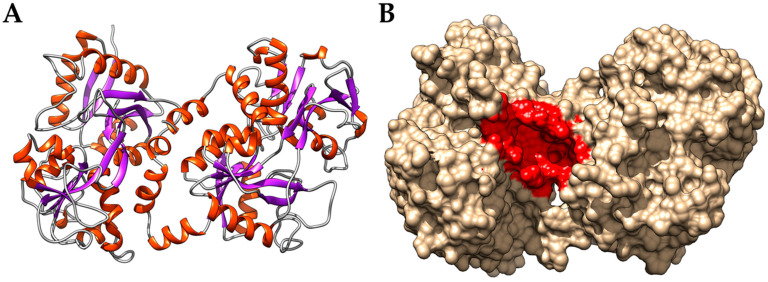
(**A**) Cartoon representation of the bovine LF structure. α-helices are represented as orange spirals, while β-strands through violet arrows. (**B**) Molecular surface representation highlighting the cavity (in red) identified in Huang et al., 2018, selected as a binding site for the molecular docking experiments.

**Figure 2 ijerph-19-05254-f002:**
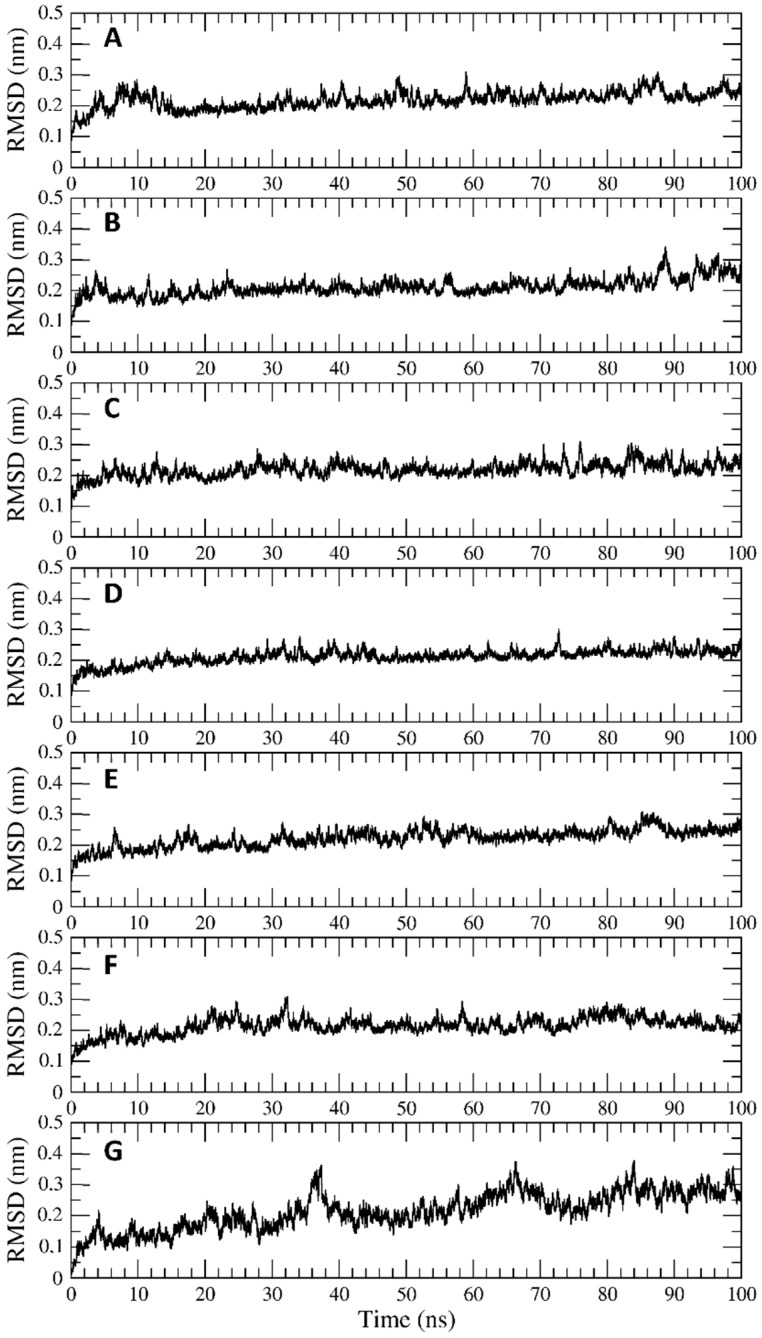
RMSD values as a function of simulation time for the complexes between bovine lactoferrin and (**A**) Epigallocatechin-3-gallate, (**B**) Taxifolin-3-glucoside, (**C**) Gallocatechin, (**D**) Artelin, (**E**) Pentagalloyl glucose, (**F**) Vitexin, (**G**) Isoorientin.

**Figure 3 ijerph-19-05254-f003:**
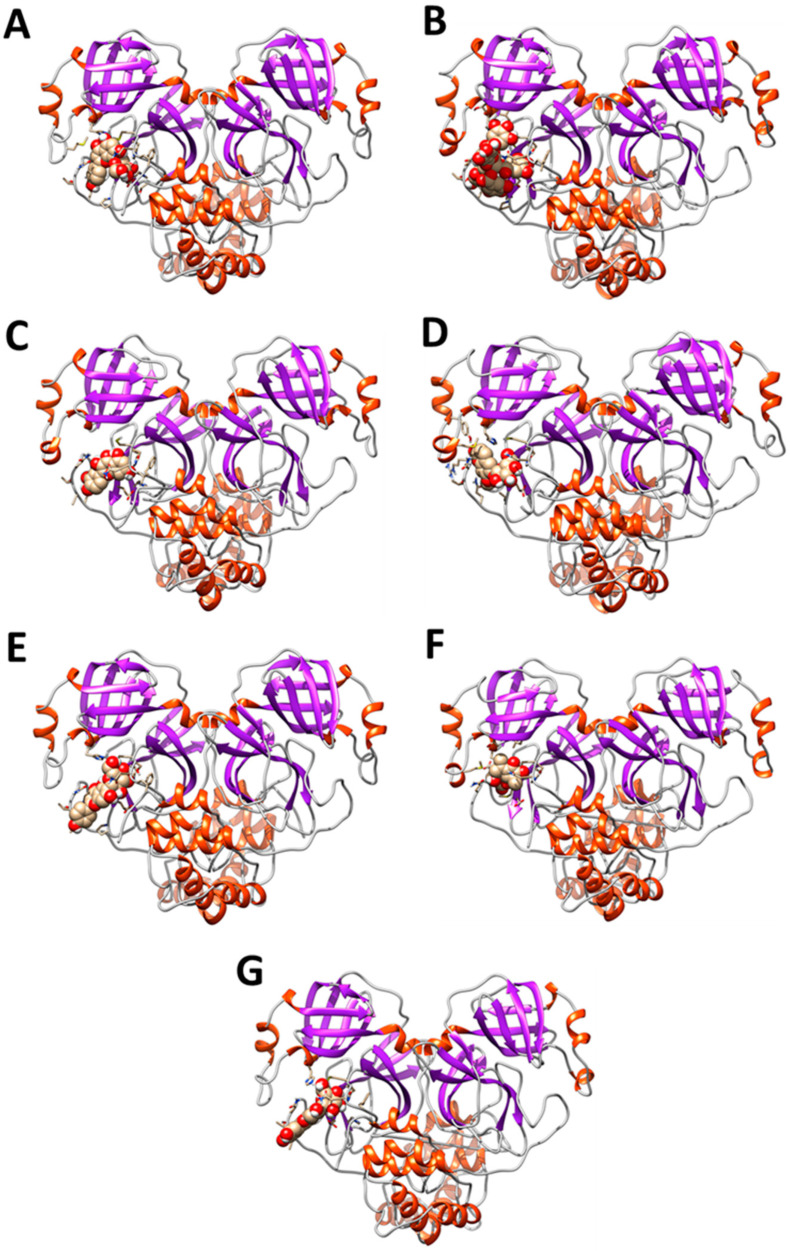
(**A**–**G**) shows the data related to the distance observed as a function of the simulation time, for the seven simulated systems, between the active site of the protease and each compound. As can be observed, the compounds identified by molecular docking remain stably anchored within the active site of the viral protease.

**Figure 4 ijerph-19-05254-f004:**
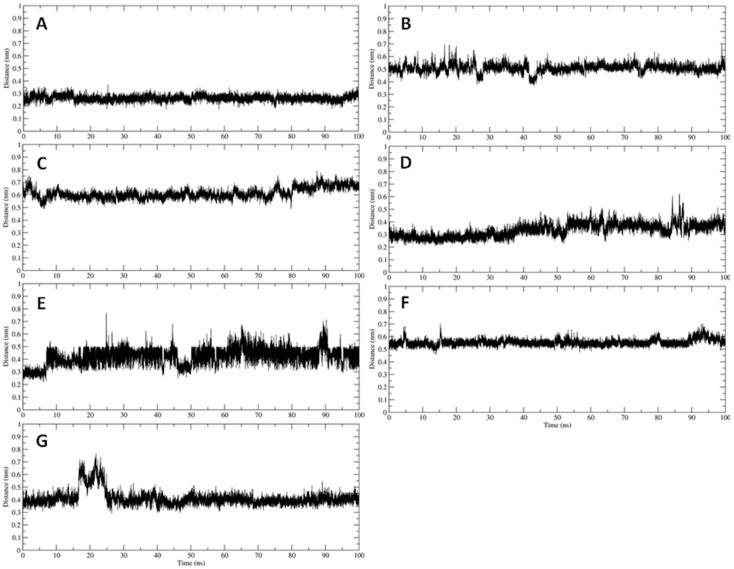
Distance as a function of simulation time between the active site of the protease and the compounds identified through docking experiments. (**A**) Complex between the protease and vitexin. (**B**) Complex between the protease and pentagalloyl-glucose. (**C**) Complex between protease and quercetin. (**D**) Complex between protease and magniolioside. (**E**) Complex between protease and isovitexin. (**F**) Complex between the protease and artelin. (**G**) Complex between the protease and the isoorientine.

**Figure 5 ijerph-19-05254-f005:**
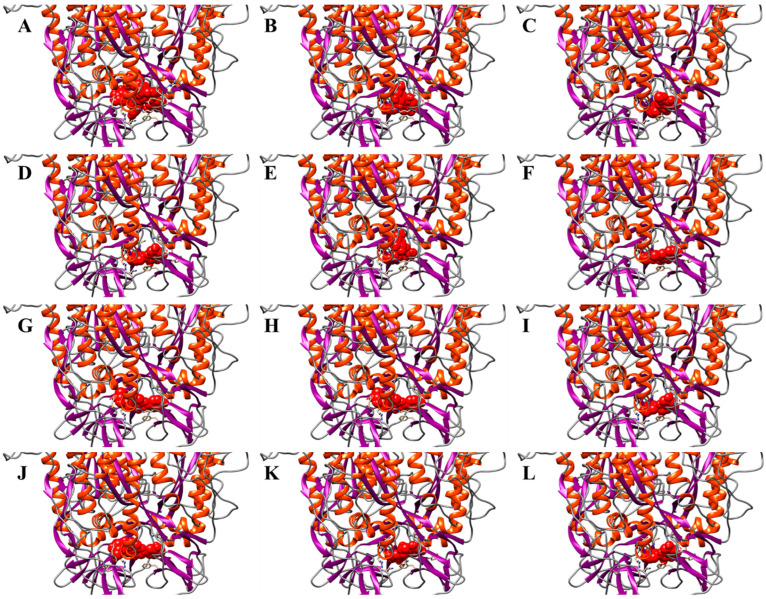
Molecular representation of the thirteen best complexes identified by docking. α-helices are represented as orange spirals, β-strands through violet arrows, while the compounds are shown as red spheres. (**A**) Complex between spike glycoprotein and Pentagalloyl glucose. (**B**) Complex between spike glycoprotein and Vitexin. (**C**) Complex between spike glycoprotein and 6-8-dihydroxy-7-methoxycoumarin. (**D**) Complex between spike glycoprotein and Quercetin. (**E**) Complex between spike glycoprotein and Taxifolin-3-glucoside. (**F**) Complex between spike glycoprotein and Isofraxoside. (**G**) Complex between spike glycoprotein and Isovitexin. (**H**) Complex between spike glycoprotein and Gallocatechin. (**I**) Complex between spike glycoprotein and Isoorientina. (**J**) Complex between spike glycoprotein and Epigallocatechin-3-gallate. (**K**) Complex between spike glycoprotein and Artelin. (**L**) Complex between the spike glycoprotein and the Magnolioside.

**Figure 6 ijerph-19-05254-f006:**
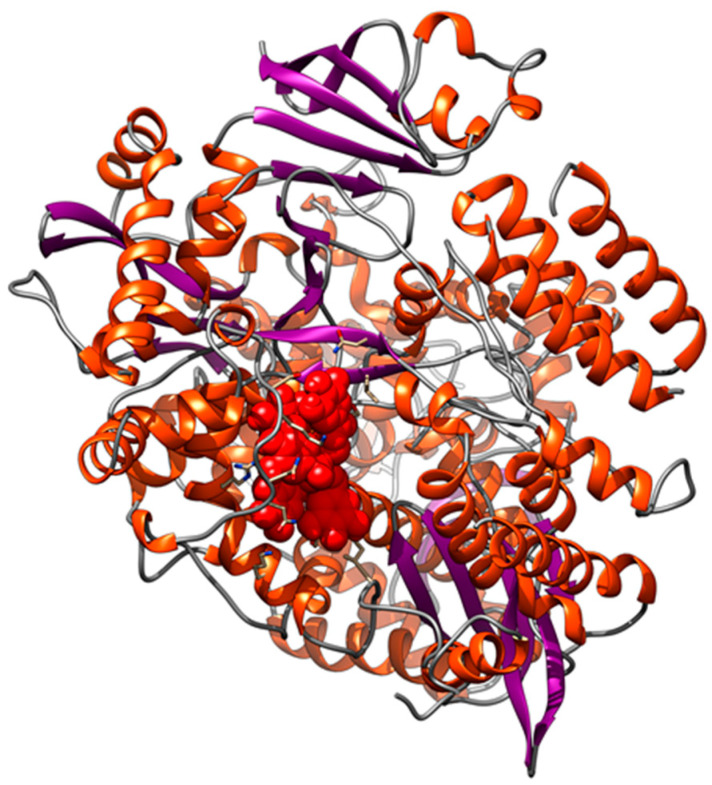
Molecular representation of the complex between Pentagalloyl glucose and the RdRp polymerase. α-helices are represented as orange spirals, β-strands through violet arrows, while the compound is shown as red spheres.

**Figure 7 ijerph-19-05254-f007:**
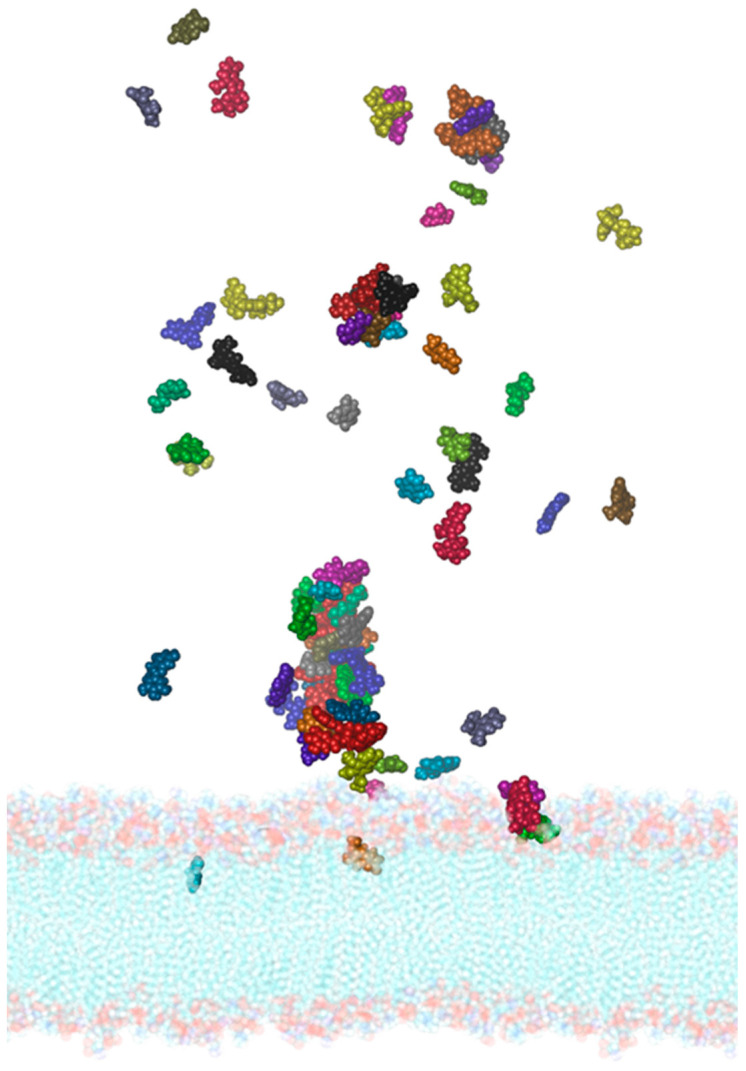
Molecular representation of the membrane-*PEL* compounds system after 150 ns of simulation time. The different colors of the spheres indicate different compounds. From the transparent side view, it is possible to observe the entry of dihydrobenzoic acid (cyan), of vanillic (gray) and homovanillic acid (purple), of scopoletin (orange), of vitexin (green), of isofraxoside (red), and gallocatechin (lilac).

**Figure 8 ijerph-19-05254-f008:**
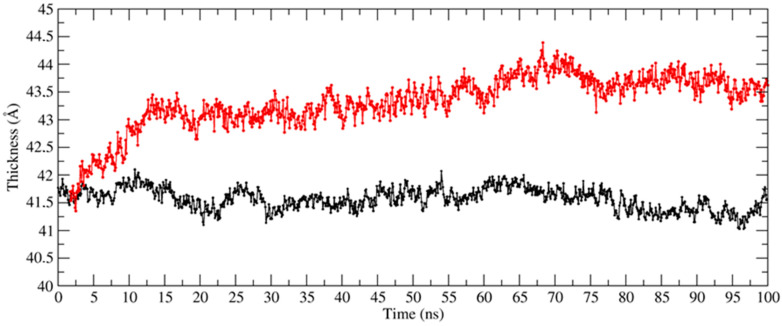
Membrane thickness as a function of simulation time for the system simulated in the absence (black) and presence (red) of the *PEL* compounds.

**Figure 9 ijerph-19-05254-f009:**
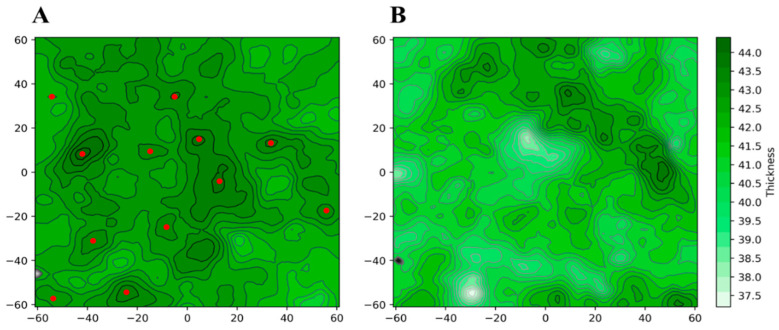
Membrane thickness as a function of the transverse plane of the membrane for the system simulated in the presence (**A**) and absence (**B**) of the *PEL* compounds.

**Table 1 ijerph-19-05254-t001:** Bovine lactoferrin molecular docking results. The average interaction energy calculated from three independent docking runs, including standard deviation, is reported for each compound.

Compound	Interaction Energy (kcal/mol ± SD)
Epigallocatechin-3-gallate	−8.1 (0.3)
Taxifolin-3-glucoside	−7.9 (0.2)
Gallocatechin	−7.7 (0.2)
Artelin	−7.7 (0.2)
Pentagalloylglucose	−7.6 (0.4)
Vitexin	−7.6 (0.3)
Isoorientin	−7.5 (0.2)
Isovitexin	−7.3 (0.2)
Orientin	−7.2 (0.3)
Magnolioside	−7.2 (0.3)
Quercetin	−7.1 (0.2)
6-8-dihydroxy-7-methoxycoumarin	−7.0 (0.3)
7-acetoxy-5-6-dimethoxycoumarin	−6.8 (0.2)
Dimethoxycoumarin	−6.4 (0.3)
Fraxetin	−6.4 (0.2)
6-7-8-trihydroxycoumarin	−6.2 (0.4)
Isofraxoside	−6.2 (0.2)
Umckalin	−6.2 (0.2)
Scopoletin	−6.0 (0.2)
5-6-7-trimethoxycoumarin	−6.0 (0.2)
Caffeic acid	−5.7 (0.3)
Ferulic acid	−5.7 (0.2)
Vanillic acid	−5.4 (0.3)
Gallic acid	−5.3 (0.2)
Apocynin	−5.3 (0.2)
Homovanillic acid	−5.3 (0.1)
Dihydroxybenzoic acid	−4.7 (0.2)

**Table 2 ijerph-19-05254-t002:** Results of MM-GBSA analysis of the bovine Lf-*PEL* compounds complexes. For each compound, we reported the average interaction energy calculated over the last 30 ns of the trajectory.

Compound	Interaction Energy (kcal/mol ± SD)
Taxifolin-3-glucoside	−39.1 (4.5)
Isoorientin	−33.7 (3.6)
Gallocatechin	−30.8 (2.8)
Artelin	−30.0 (3.4)
Vitexin	−24.6 (4.8)
Epigallocatechin-3-gallate	−21.3 (2.5)
Pentagalloyl glucose	−16.6 (4.2)

**Table 3 ijerph-19-05254-t003:** 3CLpro molecular docking results. The average interaction energy calculated from three independent docking runs, including standard deviation, is reported for each compound.

Compound	Interaction Energy (kcal/mol ± SD)
Vitexin	−8.7 (0.3)
Pentagalloylglucose	−8.3 (0.4)
Quercetin	−8.1 (0.2)
Magnolioside	−8.0 (0.3)
Isovitexin	−8.0 (0.2)
Artelin	−8.0 (0.2)
Isoorientin	−8.0 (0.2)
Orientin	−7.4 (0.3)
Taxifolin-3-glucoside	−7.4 (0.1)
Epigallocatechin-3-gallate	−7.3 (0.3)
Gallocatechin	−7.3 (0.2)
Isofraxoside	−7.2 (0.2)
Fraxetin	−6.9 (0.2)
7-acetoxy-5-6-dimethoxycoumarin	−6.7 (0.2)
Dimethoxycoumarin	−6.4 (0.3)
6-8-dihydroxy-7-methoxycoumarin	−6.2 (0.3)
Dihydroxybenzoic acid	−6.2 (0.2)
Scopoletin	−5.9 (0.2)
6-7-8-trihydroxycoumarin	−5.7 (0.4)
Caffeic acid	−5.7 (0.3)
5-6-7-trimethoxycoumarin	−5.7 (0.2)
Umckalin	−5.7 (0.2)
Ferulic acid	−5.6 (0.2)
Gallic acid	−5.6 (0.2)
Vanillic acid	−5.3 (0.3)
Apocynin	−5.3 (0.2)
Homovanillic acid	−4.6 (0.1)

**Table 4 ijerph-19-05254-t004:** Results of MM-GBSA analysis of the 3CLpro-*PEL* compounds complexes. For each compound, we reported the average interaction energy calculated over the last 30 ns of the trajectory.

Compound	Interaction Energy (kcal/mol ± SD)
Vitexin	−29.87 (3.40)
Pentagalloyl-glucose	−25.74 (5.99)
Magnolioside	−25.44 (4.05)
Isoorientin	−23.14 (3.95)
Artelin	−21.76 (5.59)
Quercetin	−15.21 (2.69)
Isovitexin	−15.08 (3.59)

**Table 5 ijerph-19-05254-t005:** Spike glycoprotein molecular docking results. The average interaction energy calculated from three independent docking runs, including standard deviation, is reported for each compound.

Compound	Interaction Energy (kcal/mol ± SD)
Pentagalloylglucose	−10.4 (0.3)
Vitexin	−9.5 (0.2)
6-8-Dihydroxy-7-methoxycoumarin	−9.3 (0.3)
Quercetin	−9.3 (0.2)
Taxifolin-3-glucoside	−9.1 (0.1)
Isofraxoside	−8.7 (0.2)
Isovitexin	−8.6 (0.2)
Gallocatechin	−8.6 (0.2)
Isoorientin	−8.5 (0.2)
Epigallocatechin-3-gallate	−8.3 (0.3)
Artelin	−8.2 (0.2)
Magnolioside	−8.0 (0.1)
Orientin	−7.4 (0.3)
Fraxetin	−7.4 (0.2)
Scopoletin	−7.0 (0.2)
7-acetoxy-5-6-dimethoxycoumarin	−6.8 (0.2)
Umckalin	−6.8 (0.2)
Caffeic acid	−6.6 (0.3)
5-6-7-trimethoxycoumarin	−6.6 (0.2)
Ferulic acid	−6.4(0.2)
Dimethoxycoumarin	−6.4 (0.3)
Homovanillic acid	−6.1 (0.1)
Gallic acid	−5.7(0.2)
6-7-8-trihydroxycoumarin	−5.7 (0.4)
Vanillic acid	−5.6 (0.3)
Apocynin	−5.3 (0.2)
Dihydroxybenzoic acid	−5.2 (0.2)

**Table 6 ijerph-19-05254-t006:** Results of MM-GBSA analysis of the Spike-*PEL* compounds complexes. For each compound, we reported the average interaction energy calculated over the last 30 ns of the trajectory.

Compound	Interaction Energy (kcal/mol ± SD)
Taxifolin-3-glucoside	−43.2 (5.8)
Pentagalloylglucose	−38.7 (5.5)
Magnolioside	−32.2 (4.2)
Isovitexin	−26.3 (4.7)
Epigallocatechin-3-gallate	−25.8 (3.4)
Gallocatechin	−24.7 (3.9)
Isofraxoside	−23.3 (4.7)
Isoorientin	−20.9 (6.8)
6-8-dihydroxy-7-methoxycoumarin	−19.4 (4.5)
Quercetin	−19.3 (4.5)
Vitexin	−18.9 (6.1)

**Table 7 ijerph-19-05254-t007:** RdRp polymersase molecular docking results. The average interaction energy calculated from three independent docking runs, including standard deviation, is reported for each compound.

Compound	Interaction Energy (kcal/mol ± SD)
Pentagalloyl glucose	−9.0 (0.5)
Vitexin	−7.4 (0.1)
Taxifolin-3-glucoside	−7.4 (0.1)
Magnolioside	−7.3 (0.2)
Isovitexin	−7.3 (0.2)
Isoorientin	−7.3 (0.3)
Quercetin	−7.2 (0.2)
Isofraxoside	−7.2 (0.2)
Epigallocatechin-3-gallate	−6.9 (0.3)
Gallocatechin	−6.9 (0.2)
Orientin	−6.8 (0.1)
Artelin	−6.4 (0.2)
6-8-dihydroxy-7-methoxycoumarin	−6.2 (0.3)
Fraxetin	−6.0 (0.1)
7-acetoxy-5-6-dimethoxycoumarin	−5.9 (0.2)
Gallic acid	−5.9 (0.2)
Caffeic acid	−5.8 (0.4)
Scopoletin	−5.8 (0.1)
6-7-8-trihydroxycoumarin	−5.7 (0.3)
Vanillic acid	−5.7 (0.3)
Umckalin	−5.7 (0.2)
Ferulic acid	−5.6 (0.2)
Dimethoxycoumarin	−5.5 (0.3)
Apocynin	−5.5 (0.2)
Dihydroxybenzoic acid	−5.4 (0.3)
5-6-7-trimethoxycoumarin	−5.4 (0.1)
Homovanillic acid	−5.4 (0.1)

## Data Availability

All data generated or analyzed during this study are included in this article. Further enquiries can be directed to the corresponding author.

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
