# Peer review of "Interaction of Pelargonium sidoides Compounds with Lactoferrin and SARS-CoV-2: Insights from Molecular Simulations"

_ijerph, 2022, doi:10.3390/ijerph19095254_

Round 1
Reviewer 1 Report
MS: Interaction of PELargonium sidoides compounds with lactofer[1]rin and SARS-CoV-2: insights from molecular simulation
The manuscript is very good and well written. There are some comments to be considered:
- I prefer to select certain plant cultivated ibn selected area and run LC/MS to determine exactly the active components and then run the docking studies on these compounds.
- Determination of the toxicity (ProTox-II webserver) for the compounds will add a value to this manuscript
- Abbreviations should be mentioned for the 1st time in the MS
- fenolic acid should be replaced by phenolic acids
- PELargonium to be corrected to Pelargonium
Author Response
Reviewer 1
MS: Interaction of Pelargonium sidoides compounds with lactoferrin and SARS-CoV-2: insights from molecular simulation
The manuscript is very good and well written. There are some comments to be considered:
We thank the referee for his positive evaluation.
I prefer to select certain plant cultivated in selected area and run LC/MS to determine exactly the active components and then run the docking studies on these compounds.
We thank the referee for his valuable comment. Indeed, the correct procedure is the one he reported; however, several papers have been already published reporting the active components present in the root extract of Pelargonium sidoides, as well as an assessment report on Pelargonium sidoides by the European Medicines Agency. Both these documents describe the compounds used for the molecular simulations carried out in our work.
Determination of the toxicity (ProTox-II webserver) for the compounds will add a value to this manuscript.
We thank the referee for his valuable comment. We added a section dedicated to the discussion of the Pelargonium sidoides compounds toxicity (Prediction of PEL compounds toxicity, red paragraph) and a Table in the supplementary material.
Abbreviations should be mentioned for the 1st time in the MS
We thank the referee for pointing this out. We corrected it in the abstract.
fenolic acid should be replaced by phenolic acids
We thank the referee for pointing this out. We corrected it all over the text.
PELargonium to be corrected to Pelargonium
We thank the referee for pointing this out. We corrected it all over the text.
Reviewer 2 Report
Current report is going to develop the product for attenuation of COVID-19 pandemic from an African medicinal plant PELargonium sidoides (PEL) and a glycoprotein of the transferrin family lactoferrin (Lf). Please conduct the concerns below.
- Docking was applied in the analysis. Reason for application of Autodock Vina 1.1.2 program needs to introduce in clear.
- The direct interaction between PEL compounds and the Lf protein and with SARS-CoV-2 Spike, 3CLPro, RdRp proteins and membrane has been indicated using the machine-aided learning. What is the benefit of this result?
- It has been established that docking is impossible to distinguish the ligand belonging agonist or antagonist. How to know the inhibition of SARS-CoV-2?
- PEL and Lf used alone were able to reduce LPS-induced proinflammatory IL-1β, as well as reduce ROS, nitrite and bacteria growth. However, these previous results are not enough to support PEL and Lf may inhibit SARS-CoV-2.
- Direct evidence shown SARS-CoV-2 is extremely required for the hypothesis.
- Interaction with the ACE2 receptor was ignored. Why?
- Specific compound or molecule in this product was not distinguished in clear.
- Novelty is not indicated in conclusion. Additionally, reproducible of the docking tool as current report is easy or not?
Author Response
Reviewer 2
Current report is going to develop the product for attenuation of COVID-19 pandemic from an African medicinal plant PELargonium sidoides (PEL) and a glycoprotein of the transferrin family lactoferrin (Lf). Please conduct the concerns below.
Docking was applied in the analysis. Reason for application of Autodock Vina 1.2.0 program needs to introduce in clear.
We thank the referee for pointing this out. There was also an error in the program version that has been fixed. Now the reason for the use of Autodock Vina 1.2.0 is stated in the text (red in Methods section).
The direct interaction between PEL compounds and the Lf protein and with SARS-CoV-2 Spike, 3CLPro, RdRp proteins and membrane has been indicated using the machine-aided learning. What is the benefit of this result?
The possibilities offered by docking in combination with molecular dynamics simulations approach are important for the identification of novel applications of known natural compounds. In this work, we have utilized these valuable tools to hypothesize interactions between PEL molecules and viral components presenting them as a preliminary suggestion, endorsed by previously published literature.
It has been established that docking is impossible to distinguish the ligand belonging agonist or antagonist. How to know the inhibition of SARS-CoV-2?
We have not considered this aspect, not detectable using these simulation methods, because we already know by literature that both lactoferrin and PEL compounds are able to reduce the viral entry into the cell (Campione et al., 2021; Papies at al., 2021).
PEL and Lf used alone were able to reduce LPS-induced proinflammatory IL-1β, as well as reduce ROS, nitrite and bacteria growth. However, these previous results are not enough to support PEL and Lf may inhibit SARS-CoV-2.
Direct evidence shown SARS-CoV-2 is extremely required for the hypothesis.
In a previous study, Terlizzi et al. (Bagnulo and Di Maio collaborated also in study) demonstrated in vitro by cytokine measurements and nitrite assay proinfiammatory activity of PEL and LF to reduce bacteria growth, whereas in several studies Campione et al., demonstrated anti-inflammatory activity of Lf against SARS-CoV-2. Papies et al., also showed activity of PEL direct against SARS-CoV-2. In this computational study we evidenced and analyzed the simultaneous capacity of PEL and Lf to interacted synergy against SARS-CoV-2, but other studies are needed. References:
Terlizzi M, Colarusso C, Di Maio U, Bagnulo A, Pinto A, Sorrentino R. Antioxidant and antimicrobial properties of Pelargonium sidoides DC and lactoferrin combination. Biosci Rep. 2020 Nov 27;40(11):BSR20203284. doi: 10.1042/BSR20203284. PMID: 33119061; PMCID: PMC7672805.
Campione E, Lanna C, Cosio T, Rosa L, Conte MP, Iacovelli F, Romeo A, Falconi M, Del Vecchio C, Franchin E et al.. Lactoferrin Against SARS-CoV-2: In Vitro and In Silico Evidences. Front Pharmacol. 2021 Jun 17;12:666600. doi: 10.3389/fphar.2021.666600. PMID: 34220505; PMCID: PMC8242182.
Campione E, Lanna C, Cosio T, Rosa L, Conte MP, Iacovelli F, Romeo A, Falconi M, Del Vecchio C, Franchin E, et al. Lactoferrin as Antiviral Treatment in COVID-19 Management: Preliminary Evidence. Int J Environ Res Public Health. 2021 Oct 19;18(20):10985. doi: 10.3390/ijerph182010985. PMID: 34682731; PMCID: PMC8535893
Papies J, Emanuel J, Heinemann N, Kulić Ž, Schroeder S, Tenner B, Lehner MD, Seifert G and Müller MA (2021) Antiviral and Immunomodulatory Effects of Pelargonium sidoides DC. Root Extract EPs® 7630 in SARS-CoV-2-Infected Human Lung Cells. Front. Pharmacol. 12:757666. doi: 10.3389/fphar.2021.757666
Interaction with the ACE2 receptor was ignored. Why?
We thank the referee for raising this question. Searching for molecules able to hamper the interaction of the Spike glycoprotein with the ACE2 receptor was not the primary goal of our study. Rather, we focused our attention on the possible effects on the viral components. In fact, Pelargonium extracts also possess antiviral activity against other types of viruses like HSV-1 and HSV-2, affecting the virus before penetration into the host cell when is pretreated with the plant extract. This suggests that PEL compounds have a direct effect on the viruses and not on the receptors they use to enter the cells. So, we did not consider ACE2 as a possible target of PEL compounds.
Specific compound or molecule in this product was not distinguished in clear.
We agree with the referee that molecules contained in Pelargonium sidoides can be found also in other traditional phytotherapics. However, unlike other vegetal species, Pelargonium sidoides extracts have well-documented therapeutic benefits against infections and aqueous-ethanolic commercial extracts like EPs® 7630 have been elaborated from the traditional herbal medicine and successfully introduced into modern phytotherapy.
Novelty is not indicated in conclusion.
Thank you for the comment. We included the judgment in the text.
Additionally, reproducible of the docking tool as current report is easy or not?
The methods section contains all the information required to reproduce the carried out molecular docking simulations, from the identity of the compounds to the AutoDock Vina run parameters.
Reviewer 3 Report
In this paper, the authors applied computational methods to check for the occurrence of interactions of the PELargonium sidoides (PEL ) compounds with lactoferrin (Lf) and SARS-CoV-2 components.
Specifically, they evaluated a possible interaction between PEL compounds and the SARS-CoV-2 3CLpro protein, with SARS-CoV-2 Spike glycoprotein and with SARS-CoV-2 RdRp polymerase and verified whether the PEL compounds could interact with the membrane of the SARS-CoV-2 virus. Moreover, they research for possible direct interaction between PEL compounds and Lf as in literature is reported that combination PELargonium sidoides+Lf can reduce in vitro the release of pro-inflammatory cytokines, oxidants, and bacteria growth.
With computational studies, they found that the structure and activity of lactoferrin is not altered by the presence of organic molecules as PEL compounds or metal ions different from iron. However, in my opinion, they cannot affirm that “Based on these results, the combined use of PEL and Lf in a dietary supplement has been acknowledged”, becouse other experiments are needed for a statement like that. So, this sentence should be adapted.
The article seems well written and well executed, with a large methodological part.
However, please consider the following suggestions:
-Insert the chemical structures of the PEL compounds or at least those that are more active from the studies carried out.
-If possible, for the compounds found most interesting in the experiments conducted, insert a table with their quantities (i.e. %) within the PEL extract to better understand the order of magnitude of these compounds within the PEL extract.
-Although it is clear from the bibliographic references cited, please specify in this article the origin of the PEL extract (i.e.root )
-Insert the references to support that PEL and Lf could synergistically interfering in vivo (see “conclusion” paragraph ) with the mechanism of infection of SARS-CoV-2 or remodulate this sentence.
Author Response
Reviewer 3
In this paper, the authors applied computational methods to check for the occurrence of interactions of the PELargonium sidoides (PEL) compounds with lactoferrin (Lf) and SARS-CoV-2 components.
Specifically, they evaluated a possible interaction between PEL compounds and the SARS-CoV-2 3CLpro protein, with SARS-CoV-2 Spike glycoprotein and with SARS-CoV-2 RdRp polymerase and verified whether the PEL compounds could interact with the membrane of the SARS-CoV-2 virus. Moreover, they research for possible direct interaction between PEL compounds and Lf as in literature is reported that combination PELargonium sidoides+Lf can reduce in vitro the release of pro-inflammatory cytokines, oxidants, and bacteria growth.
With computational studies, they found that the structure and activity of lactoferrin is not altered by the presence of organic molecules as PEL compounds or metal ions different from iron. However, in my opinion, they cannot affirm that “Based on these results, the combined use of PEL and Lf in a dietary supplement has been acknowledged”, becouse other experiments are needed for a statement like that. So, this sentence should be adapted.
The article seems well written and well executed, with a large methodological part.
We thank the referee for his positive evaluation.
However, please consider the following suggestions: Insert the chemical structures of the PEL compounds or at least those that are more active from the studies carried out.
We thank the referee for his valuable suggestion. We added a figure in the supplementary material with the chemical structure of all the simulated PEL compounds.
If possible, for the compounds found most interesting in the experiments conducted, insert a table with their quantities (i.e. %) within the PEL extract to better understand the order of magnitude of these compounds within the PEL extract.
EPs 7630 is the most common commercial and well-characterized ethanolic (11% (m/m)) extract of Pelargonium sidoides roots. Schötz et al. (2008) gave an account of the constituents of EPs 7630. Six main groups of compounds can be found: purine derivatives (2%), coumarins (2%), peptides (10%), carbohydrates (12%), minerals (12%) and oligomeric prodelphinidines (40%). However, there are no literature data describing in detail the percentages of the individual components of Pelargonium sidoides, therefore we are not able to include them in the text.
Although it is clear from the bibliographic references cited, please specify in this article the origin of the PEL extract (i.e.root )
We thank the referee for pointing this out. We specified the origin of the PEL extracts (as correctly supposed by the referee) in the text.
Insert the references to support that PEL and Lf could synergistically interfering in vivo (see “conclusion” paragraph) with the mechanism of infection of SARS-CoV-2 or remodulate this sentence.
This is a preliminary computational in silico study, where we hypothesized and validate model of interactions between the two molecules (PEL and Lf) and against SARS-CoV-2, and after this first important and necessary part we will set up in vitro and in vivo experiments. See at conclusion paragraph.
Round 2
Reviewer 2 Report
It has been revised following the comments.